# Detection of SARS-CoV-2 from Saliva as Compared to Nasopharyngeal Swabs in Outpatients

**DOI:** 10.3390/v12111314

**Published:** 2020-11-17

**Authors:** Christopher Kandel, Jennifer Zheng, Janine McCready, Mihaela Anca Serbanescu, Hilary Racher, Melissa Desaulnier, Jeff E Powis, Kyle Vojdani, Laura Finlay, Elena Sheldrake, Christie Vermeiren, Kevin Katz, Allison McGeer, Robert Kozak, Lee W Goneau

**Affiliations:** 1Mount Sinai Hospital, Toronto, ON M5G 1X5, Canada; christopher.kandel@one-mail.on.ca (C.K.); Allison.mcgeer@sinaihealth.ca (A.M.); 2Department of Medicine, University of Toronto, Toronto, ON M5S 1A8, Canada; jenniferj.zheng@mail.utoronto.ca; 3Department of Medicine, Michael Garron Hospital, Toronto, ON M4C 3E7, Canada; janine.mccready@tehn.ca (J.M.); jeff.powis@tehn.ca (J.E.P.); kyle.vojdani@tehn.ca (K.V.); laura.finlay@tehn.ca (L.F.); 4Dynacare Laboratory, Brampton, ON L6T 5M3, Canada; serbanescum@dynacare.ca (M.A.S.); racherh@dynacare.ca (H.R.); desaulnierm@dynacare.ca (M.D.); 5Department of Laboratory Medicine, University of Toronto, Toronto, ON M5S 1A8, Canada; kevin.katz@nygh.on.ca (K.K.); rob.kozak@sunnybrook.ca (R.K.); 6Shared Hospital Laboratory, Toronto, ON M4N 3M5, Canada; elena.sheldrake@nygh.on.ca (E.S.); cvermeiren@shn.ca (C.V.); 7North York General Hospital, Toronto, ON M2K 1E1, Canada; 8Sunnybrook Health Sciences Centre, Toronto, ON M4N 3M5, Canada

**Keywords:** COVID-19, SARS-CoV-2, saliva, nasopharyngeal swab

## Abstract

Widely available and easily accessible testing for COVID-19 is a cornerstone of pandemic containment strategies. Nasopharyngeal swabs (NPS) are the currently accepted standard for sample collection but are limited by their need for collection devices and sampling by trained healthcare professionals. The aim of this study was to compare the performance of saliva to NPS in an outpatient setting. This was a prospective study conducted at three centers, which compared the performance of saliva and NPS samples collected at the time of assessment center visit. Samples were tested by real-time reverse transcription polymerase chain reaction and sensitivity and overall agreement determined between saliva and NPS. Clinical data was abstracted by chart review for select study participants. Of the 432 paired samples, 46 were positive for SARS-CoV-2, with seven discordant observed between the two sample types (four individuals testing positive only by NPS and three by saliva only). The observed agreement was 98.4% (kappa coefficient 0.91) and a composite reference standard demonstrated sensitivity of 0.91 and 0.93 for saliva and NPS samples, respectively. On average, the Ct values obtained from saliva as compared to NPS were higher by 2.76. This study demonstrates that saliva performs comparably to NPS for the detection of SARS-CoV-2. Saliva was simple to collect, did not require transport media, and could be tested with equipment readily available at most laboratories. The use of saliva as an acceptable alternative to NPS could support the use of widespread surveillance testing for SARS-CoV-2.

## 1. Introduction

Severe acute respiratory syndrome coronavirus 2 (SARS-CoV-2) has infected over 32 million individuals since December 2019. Containment strategies are predicated on rapid diagnosis [1] with real-time reverse transcription polymerase chain reaction (rRT-PCR) on nasopharyngeal swab (NPS) specimens as the currently accepted gold standard [2]. However, NPS collection is uncomfortable and may result in testing hesitancy [3], specimen retrieval requires a trained healthcare professional wearing personal protective equipment, and the demand for swabs and transport media has led to shortages. 

Using saliva avoids the need for swabs and transport media and permits self-collection [4]. Saliva also remains stable for days at various temperatures [5]. However, reported sensitivity of saliva compared to NPS ranges from 69.2% to 97.6% [6,7,8,9,10]. Most studies are small and are of hospitalized patients only, and some have not investigated the possibility that saliva may identify individuals who falsely test negative by NPS. In addition, patient population, severity of symptoms, testing timing relatively to symptom onset, method of saliva collection and testing platform may all affect test performance characteristics. Thus, on-going comparisons are needed. We assessed the performance characteristics of saliva and NPS for the detection of SARS-CoV-2 in three outpatient testing centers.

## 2. Materials and Methods 

### 2.1. Study Population and Saliva Collection

All adults (age > 18) who presented to three assessment centers in Toronto, Ontario (North York General Hospital, Michael Garron Hospital, or the Occupation Health Clinic at Sunnybrook Health Sciences Centre), Canada, and had an NPS obtained for SARS-CoV-2 testing from 1 May 2020 until 10 August 2020 were asked to provide as much saliva as they could produce, up to a maximum of 5 mL, into a sterile container that was stored at 4 °C until tested (see Appendix A). Self-collection procedures are outlined in the supplementary methods. Once NPS swab results were available, all those positive for SARS-CoV-2 and a random sample of negatives from the same day (ratio of negative to positive 10:1) were selected and matched saliva specimens identified and tested. Clinical information was abstracted by chart review. This study was deemed to be a quality improvement project by the Research Ethics Board of participating hospitals.

### 2.2. SARS-CoV-2 RNA Detection

NPS samples were tested at Shared Hospital Laboratory (Toronto, ON, Canada). Samples were collected in either universal transport media (UTM; Copan, Italy) or Liquid Amies solution (Eswab collection, Copan, Italy) and nucleic acids using Promega Maxwell HT Viral TNA Kit. Samples were extracted and prepared for PCR using the EP motion 5075 liquid handler (Eppendorf, Mississauga, ON, Canada) with 250 uL of patient material being extracted using the TNA viral extraction kit (Promega) with a final elution volume of 30 µL. Detection of SARS-CoV-2 E-gene and 5′-UTR was performed on the CFX96 Touch Real-time PCR detection system (BioRad, Canada) [11,12]. Alternatively, samples were extracted, and nucleic acid amplification was performed on the BD MAX^TM^ system using the ExK TNA-2 strip (Becton, Dickinson, ND, USA).

Saliva samples were tested at Dynacare Laboratory (Brampton, ON, Canada) using the Roche cobas^®^ 6800 analyzer and Roche cobas^®^ SARS-CoV-2 assay (Hoffmann-La Roche Limited, Mississauga, ON, Canada). All positive samples and a random subset of 23 negative samples were also tested using a second method (see Appendix A).

Prior to testing, saliva samples were equilibrated to room-temperature and vortexed for five seconds to homogenize the sample and disrupt mucus clots. A 600 μL aliquot of each specimen was transferred to a 13 × 75 mm Starstedt tube containing 300 μL of cobas omni Lysis Reagent. In cases where the samples had less than 600 μL volume, 600 μL of cobas omni Lysis Reagent were added directly into the primary sample tube to capture any SARS-CoV-2 RNA potentially present. Samples were vortexed for five seconds and incubated at room-temperature for 10 min in the cobas omni Lysis Reagent, followed by centrifugation at 2500 RCF for 15 min at room-temperature. The samples were loaded onto the Roche cobas 6800 analyzer for testing. The cobas^®^ SARS-CoV-2 test provides fully automated sample preparation (nucleic acid extraction and amplification) and RT-PCR-based qualitative detection of SARS-CoV-2 RNA through amplification of two genomic target regions, ORF1ab and E gene. An armored RNA internal control is added to all samples to validate each reaction. Automated data management software assigns test results for all tests. This test is Health Canada approved for in vitro diagnostic use. Staff who performed saliva testing were blinded to the matched NPS result.

Stability of SARS-CoV-2 in various environmental conditions was assessed using saliva samples collected from nine volunteers. Residual SARS-CoV-2 positive patient sample was spiked into donor saliva and stored for up to seven days at room temperature, 2–8 °C, and −20 °C. Testing occurred at baseline and at days three and seven using the Roche cobas SARS-CoV-2 assay.

The limit of detection for SARS-CoV-2 RNA was determined for the Roche cobas SARS-CoV-2 assay by diluting a 4000 copies/mL positive control (Proceedx Standard, Mississauga, Ontario, Canada Microbix) into pooled negative saliva sample matrix to concentrations of 125, 62.5, 31.25, and 15.63 copies/mL. All dilutions were tested in replicates of 20.

### 2.3. Supplemental SARS-CoV-2 Testing of Positive Saliva Samples with Paired Negative NPS

All positive samples and a random subset of 23 negative samples were also tested using a second method different than Roche cobas SARS-CoV-2 assay used for primary characterization. This method involved initial offboard virus inactivation by combining 200 μL of each patient saliva sample with 250 μL of lysis buffer master mix, containing TNA lysis buffer (Omega Bio-tek, Inc., Norcoss, GA, USA), Carrier RNA (Omega Bio-tek) and MS2 phage internal control (Thermo Fisher, Waltham, MA, USA). RNA was extracted using MagBind Viral RNA Xpress kit (Omega Bio-tek) on Hamilton Microlab STARlets (Hamilton Company, Reno, NV, USA). A 10 μL aliquot of RNA was added in a 15 μL reaction using TaqPath COVID-19 Combo Kit (Thermo Fisher Scientific) on real-time PCR systems (Applied Biosystems 7500 Fast or QuantStudio 6); one cycle at 25 °C for two minutes, one cycle at 53 °C for 10 min, one cycle at 95 °C for two minutes, 40 cycles at 95 °C for three seconds and 60 °C for 30 s. A sample was defined as positive if the viral genome was detected at threshold cycle (Ct) values of <37 and as negative at Ct values ≥37.

### 2.4. Statistical Analysis

Descriptive statistics were presented as means or medians as appropriate for continuous variables and proportions for categorical variables. The kappa coefficient was used to estimate the agreement between NPS and saliva SARS-CoV-2 detection results and a paired t-test was used to compare the cycle threshold values. The sensitivity for NPS and saliva were calculated using the total number of positive specimens by either test as the reference standard. All analyses were performed using R, version 4.0.0 (R statistical computing). 

## 3. Results

Overall, 432 paired saliva/NPS patient samples were tested and included in final analysis: 253 (59%), 108 (25%), and 71 (16%) from sites A, B, and C, respectively. Ten (2.3%) of the saliva specimens were highly mucoid; and 54 (13%) were small volume (<0.5 mL) requiring the addition of lysis buffer to a total volume of at least 0.5 mL for testing. Thirty-eight (8.80%) were “invalid” on initial testing, with three (0.69%) yielding repeated invalid results on re-testing. These three repeatedly invalid samples were excluded from final analysis. All invalid results obtained in this study were due to clot errors detected on the Roche cobas 6800 during sample aspiration. The median time from saliva collection to testing was six days (interquartile range (IQR) 5 to 8). 

All patients were tested at out-patient assessment centers. Testing characteristics were available from site A only for 236 (54.5%) patients who provided a saliva specimen. The median age of persons with COVID-19 was 42 (IQR 30 to 54), 134 (56.8%) were female, 81 (34.3%) were asymptomatic, and in those with symptoms the median time from symptom onset to testing was 4 days (IQR 2–7 days). Only one individual later required hospitalization. 

Seven of 432 samples were discordant between the two sample types, with four individuals testing positive only by NPS and three by saliva only (Table 1 and Table 2), and an observed agreement was 98.4% (kappa coefficient = 0.91, 95% CI 0.82–0.96). 

The sensitivity of NPS was 0.93 (95% CI 0.81–0.99) and that of saliva 0.91 (95% CI 0.79–0.98). The rRT-PCR cycle threshold results for concordantly positive NPS and saliva specimens were available for 33/39 pairs (Figure 1). Cycle threshold values obtained from saliva were higher than those from NPS (median difference in Ct 2.76; 95% CI 0.36–5.15, *p* = 0.03). The limit of detection for SARS-CoV-2 in saliva was 31.25 copies/mL using the Roche cobas SARS-CoV-2 assay.

Viral RNA stability was evaluated by storing spiked donor saliva samples at different temperatures (4 °C, room temperature, and −20 °C) and testing at days 3 and 7. There was no significant difference in Ct values for the ORF1 or E genes in samples tested on day 3 and day 7 compared to initial inoculation (day 0) for samples stored at room temperature or at 4 °C. The median differences for Ct for E gene at room temperature and 4 °C between day 7 and day 0 was −0.3 and −0.8, *p* = 1.0 and 0.64, respectively (Table 3). Samples stored at −20 °C had higher Ct values on day 7 compared to day 0 (median 1.4 higher, range −0.2 to +3.0, *p* = 0.01).

## 4. Discussion

In our out-patient assessment centers, using saliva to diagnose COVID-19 compared favorably to NPS. Saliva was simple to self-collect, needed minimal instruction, avoided the presence of a healthcare professional, and was stable for seven days. 

We found that the sensitivity for detecting SARS-CoV-2 in saliva is 91% (95% CI 79–98%). This is comparable to a recent meta-analysis which reported a pooled sensitivity of 0.91 (95% CI 0.80–0.99) [13], and not different from NPS. The limit of detection for SARS-CoV-2 in saliva was also consistent with NPS-tested samples as reported by Roche (31.25 copies/mL and 25–32 copies/mL for saliva and NPS, respectively). Although an increased average Ct value was observed in saliva compared to NPS, this difference was not significant. Interestingly, decreased Ct was observed in some donor samples during sample stability studies, suggesting a paradoxical increase in SARS-CoV-2 RNA. This effect has been observed in other evaluations of SARS-CoV-2 stability in saliva and is thought to be due to assay variability or possibly even virus replication in residual host cells within the sample [5]; however, this effect was not explored further as part of this study.

Many studies are predicated on the assumption that the NPS is the reference standard, which is imperfect given the challenges in obtaining a correct specimen and the possible lower viral loads present in the nasopharynx [9]. As such, a strength of this study, beyond the number of positive specimens tested, is the inclusion of a large number of individuals who tested negative by NPS. Using this approach, we were able to demonstrate that SARS-CoV-2 is detectable in patients who tested negative in a matched NPS sample. The somewhat higher Ct values for saliva compared to NPS in our study is consistent with the somewhat lower sensitivity of saliva compared to NPS later in illness when viral loads in the airway are lower. 

Several studies have demonstrated the efficacy of testing saliva for the detection of SARS-CoV-2, however the majority of these used matched samples from hospitalized patients [9,14,15,16], or use commercial saliva collection kits with transport media [8]. Variability in the performance characteristics of saliva may be explained by differing intervals for patient sampling between studies. Extended duration between the onset of symptoms and testing saliva can negatively impact detection rates [17]. However, recent findings by Wyllie and colleagues demonstrated a higher percentage of positive saliva samples compared to NPS in a hospitalized cohort [9]. Differing testing methods and sample preparation strategies are likely to contribute to variable performance [14,18], while individuals presenting to an outpatient setting often differ from inpatients with less severe symptoms that can translate into lower viral loads [19]. Our study population is representative of outpatients with almost 30% being asymptomatic. However, there is still a lack of consensus that disease severity correlates with viral load as similar studies comparing symptomatic and asymptomatic patients have demonstrated similar Ct values [20,21]. Patient demographic or clinical presentation at the time of sample collection did not appear to impact test outcome between sample types in this study.

A potential drawback of testing saliva is the differing viscosity and consistency of the specimens received by the microbiology laboratory. Samples with insufficient volume were frequently encountered, though this did not appear to impact sensitivity. Preparation of saliva was also more laborious compared to NPS, with highly mucoid samples requiring additional homogenizing to prevent instrument clotting failures. Although the invalid rate for saliva was improved from 8.80% to 0.69% through centrifugation, this is still much higher than the invalid rate for NPS collected samples, which is 0.04%. Importantly, one discordant saliva sample that was falsely-negative compared to the matched NPS-collected sample was noted to be highly mucoid and challenging to pipette, requiring vigorous vortexing.

There are limitations of the study that merit emphasis. First, not all individuals who tested positive by NPS agreed to submit a saliva specimen for testing. This may simply be because submission of saliva was voluntary, but we cannot exclude the possibility that some participants may not have been able to produce saliva on demand. Second, our sample cohort was adults, and our results may not apply to children. Third, in our low prevalence setting, false positive results may occur. However, testing for two targets and the low cycle threshold values observed make it likely that all were truly positive. Fourth, different assays were used for testing of saliva compared to NPS, which is a potential cause of the differences in Ct values.

## 5. Conclusions

In conclusion, saliva is an acceptable alternative to an NPS for the detection of SARS-CoV-2 in adult outpatients. The simple and non-invasive self-collection procedure avoids the need for a swab, reduces exposure to healthcare workers, and can be self-collected in any environment. This has the potential to be used in settings such as workplace screening, where repeated testing may be needed, or in outbreak settings where testing of large numbers of people simultaneously is warranted.

## Figures and Tables

**Figure 1 viruses-12-01314-f001:**
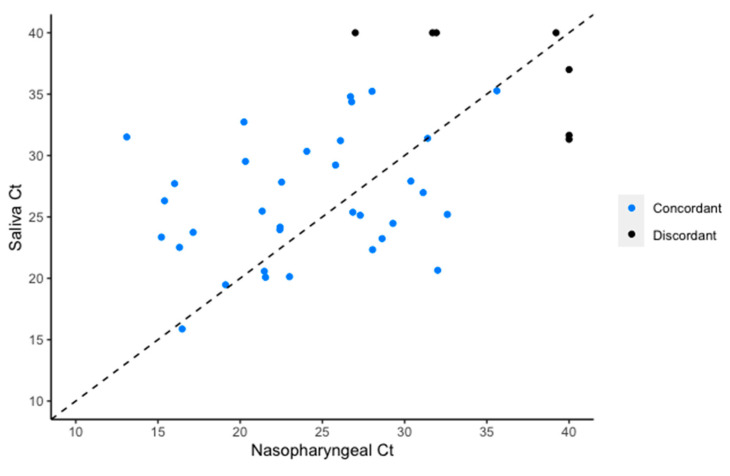
Cycle threshold values for the E gene target of SARS-CoV-2 RNA retrieved from individuals who were concordantly positive by nasopharyngeal swab and saliva.

**Table 1 viruses-12-01314-t001:** Results of SARS-CoV-2 detection in paired nasopharyngeal swab and saliva specimens from individuals who presented to an outpatient testing center.

Saliva Sample	Nasopharyngeal Swab		Total
	Positive	Negative	
Positive	39	3	42
Negative	4	383	387
Total	43	386	429

**Table 2 viruses-12-01314-t002:** Discordant results of matched nasopharyngeal swab (NPS) and saliva sample pairs.

	Results		NPS C_T_ Values			Saliva C_T_ Values		
	NPS	Saliva	Target 1 *	Target 2 ^†^	Internal Control	Target 1 ^‡^	Target 2 ^§^	Internal Control
Sample 1	positive	negative	27.0	26.8	25.1	ND	ND	35.3
Sample 2	positive	negative	39.2	38.9	26.1	ND	ND	35.0
Sample 3	positive	negative	31.7	33.5	21.2	ND	ND	34.5
Sample 4	positive	negative	31.9	31.8	20.7	ND	ND	22.5
Sample 5	negative	positive	ND	ND	21.5	ND	37.0	36.8
Sample 6	negative	positive	ND	ND	24.2	29.4	31.7	38.6
Sample 7	negative	positive	ND	ND	27.6	30.1	31.3	36.2

* Target 1: E-gene; ^†^ Target 2: 5′-UTR; ^‡^ Target 1: ORF1ab; ^§^ Target 2: E gene; ND: not detected.

**Table 3 viruses-12-01314-t003:** Stability of SARS-CoV-2 in spiked saliva samples.

**Room Temperature (18–24 ℃)**													
**Saliva Sample**	**Day 0 (Ct)**			**Day 3 (Ct)**			**Day7 (Ct)**			**Day 3 Δ (Ct)**		**Day 7 Δ (Ct)**	
	**ORF1a/b**	**E gene**	**Internal Control**	**ORF1a/b**	**E gene**	**Internal Control**	**ORF1a/b**	**E gene**	**Internal Control**	**ORF1a/b**	**E gene**	**ORF1a/b**	**E gene**
S01	24.4	25.2	36.1	24.9	25.7	37.0	25.5	26.4	36.9	0.5	0.4	1.1	1.2
S02	25.3	26.0	36.9	24.0	24.7	36.2	24.5	25.5	37.1	−1.4	−1.3	−0.8	−0.6
S03	24.2	25.0	35.7	21.8	22.7	34.0	22.6	23.3	33.1	−2.4	−2.3	−1.6	−1.7
S04	23.4	24.1	35.2	22.4	23.3	35.9	23.9	24.6	36.1	−1.0	−0.8	0.5	0.5
S05	24.7	25.5	36.1	23.1	23.9	36.1	23.1	23.8	35.2	−1.7	−1.7	−1.7	−1.7
S06	24.5	25.3	37.1	24.4	25.1	37.1	26.6	27.3	36.8	−0.2	−0.1	2.1	2.0
S07	23.4	23.8	36.0	22.9	23.8	34.1	21.9	22.5	33.8	−0.5	0.0	−1.5	−1.3
S08	23.5	24.4	35.5	24.0	24.9	35.2	24.4	25.2	34.8	0.4	0.5	0.9	0.8
S09	23.6	24.1	34.5	22.2	22.9	33.6	22.6	23.5	33.4	−1.5	−1.2	−1.0	−0.6
**Refrigerated (2–8 ℃)**													
**Saliva Sample**	**Day 0**			**Day3**			**Day7**			**Day 3 Δ**		**Day 7 Δ**	
	**ORF1a/b**	**E gene**	**Internal Control**	**ORF1a/b**	**E gene**	**Internal Control**	**ORF1a/b**	**E gene**	**Internal Control**	**ORF1a/b**	**E gene**	**ORF1a/b**	**E gene**
S01	24.6	25.4	36.8	24.7	25.5	35.7	26.4	27.1	36.3	0.2	0.1	1.8	1.7
S02	25.6	26.3	38.7	22.8	23.8	34.2	25.3	26.0	36.9	−2.8	−2.5	−0.3	−0.3
S03	24.0	24.6	35.6	24.0	24.9	35.6	23.1	23.9	35.1	0.1	0.3	−0.9	−0.7
S04	23.8	24.4	37.7	23.5	24.3	34.6	22.9	23.7	35.7	−0.3	−0.1	−0.8	−0.7
S05	25.1	25.9	35.6	24.6	25.3	36.5	24.4	25.1	36.6	−0.5	−0.6	−0.7	−0.8
S06	24.1	25.0	36.1	25.3	26.0	36.6	25.7	26.6	37.7	1.2	1.0	1.6	1.6
S07	25.6	26.1	35.1	23.2	23.8	35.0	23.2	23.8	34.7	−2.4	−2.3	−2.4	−2.3
S08	25.5	26.5	36.2	24.5	25.4	35.0	26.2	26.9	36.7	−1.0	−1.0	0.7	0.4
S09	23.3	23.9	35.1	23.5	24.2	37.7	23.6	24.4	33.9	0.1	0.4	0.3	0.5
**Frozen (−20 ℃)**													
**Day 0** **ORF1a/b**	**Day3**			**Day7**			**Day 3 Δ**			**Day 7 Δ**			
	**E gene**	**Internal Control**	**ORF1a/b**	**E gene**	**Internal Control**	**ORF1a/b**	**E gene**	**Internal Control**	**ORF1a/b**	**E gene**	**ORF1a/b**	**E gene**	
S01	24.6	25.5	36.3	24.2	25.1	36.3	26.0	26.7	35.9	−0.4	−0.3	1.4	1.3
S02	25.4	26.2	35.4	25.4	26.2	38.1	25.7	26.5	36.9	0.0	0.0	0.3	0.3
S03	25.6	26.5	35.1	25.1	26.1	34.7	25.8	26.7	35.2	−0.5	−0.3	0.2	0.2
S04	23.9	24.6	35.4	23.2	24.0	36.0	23.6	24.4	34.4	−0.7	−0.6	−0.3	−0.2
S05	25.6	26.4	36.9	23.9	24.6	36.2	27.5	28.4	36.7	−1.7	−1.8	1.9	2.0
S06	24.7	25.4	36.1	23.6	24.5	35.8	27.2	28.1	36.1	−1.1	−0.9	2.6	2.8
S07	24.8	25.3	35.3	24.5	25.2	34.3	26.9	27.7	36.1	−0.3	−0.2	2.1	2.4
S08	24.7	25.7	35.1	24.8	25.9	36.7	27.6	28.6	36.1	0.1	0.2	2.9	3.0
S09	24.2	24.7	35.9	22.7	23.6	34.1	25.1	26.1	34.1	−1.4	−1.1	1.0	1.4

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
