# Peer review of "Detection of SARS-CoV-2 from Saliva as Compared to Nasopharyngeal Swabs in Outpatients"

_viruses, 2020, doi:10.3390/v12111314_

Round 1

Reviewer 1 Report

- Line 121: “Thirty-eight (8.80%) were ‘invalid’ on initial testing, with three (0.69%) yielding repeated invalid results on re-testing.” Later on line 189, it says that the invalid rate is reduced to 0.3%, so is it 0.69% or 0.3%? Also please be more specific about what ‘invalid’ means in this context, is it always instrument clotting failures?

- For the seven discordant samples, was there any common features for the individuals or samples? For example, were these more likely to be from asymptomatic carriers, or people with later symptom onset, or low-volume saliva samples, etc? Some additional descriptions may reveal reasons that these samples are discordant, or if there is no consistent association between sample data and the discordant samples, then that is useful information for the readers as well.

- Figure 1 may be easier to read as an X-Y scatterplot, with Ct swab on the X axis and Ct Saliva on the X axis, and a diagonal line where they are equal from the bottom left to top right. The Ct cutoffs for positivity could also be indicated on such a plot, as a vertical and horizontal line across the plot space.

Currently Figure 1 only contains the concordantly positive samples, but the discordant samples should also be plotted on such a scatterplot, and can be indicated with a different color of point. Please also include the negative samples, with NAs plotted at the maximum values on the plot. This will help to evaluate the extent of difference between positive and negative samples using the two different assays, which is important for evaluating the reliability of the saliva approach.

- Were any samples tested repeatedly on the same day (technical replicates)? This would serve as a valuable baseline to evaluate the delta values in Table 3, to see if the delta Ct values are within the range expected for technical replicates. It seems as if there is some amount of technical noise, since in many cases, there is more amplification following storing for 3 or 7 days.

Author Response

  1. Line 121: “Thirty-eight (8.80%) were ‘invalid’ on initial testing, with three (0.69%) yielding repeated invalid results on re-testing.” Later on line 189, it says that the invalid rate is reduced to 0.3%, so is it 0.69% or 0.3%? Also please be more specific about what ‘invalid’ means in this context, is it always instrument clotting failures?

Thank you for highlighting this discrepancy. The correct value is 0.69%. This has been corrected on line 208 of the revised manuscript.

We have also amended the manuscript to include a statement specifying the nature of invalid results: “All invalid results obtained in this study were due to clot errors detected on the Roche cobas 6800 during sample aspiration.” – Line 128-129

  1. For the seven discordant samples, was there any common features for the individuals or samples? For example, were these more likely to be from asymptomatic carriers, or people with later symptom onset, or low-volume saliva samples, etc? Some additional descriptions may reveal reasons that these samples are discordant, or if there is no consistent association between sample data and the discordant samples, then that is useful information for the readers as well.

This has been recommendation has been addressed in response to Reviewer 2’s question number four. Specifically, there was one highly viscous saliva sample that was falsely-negative when compared to NPS testing. There were no other notable differences between paired samples discordant for SARS-CoV-2 detection in terms of clinical presentation or time from collection to testing. We have amended the manuscript to include a statement regarding the single false-negative saliva that was highly viscous: “Importantly, one discordant saliva sample that was falsely-negative compared to the matched NPS-collected sample was noted to be highly mucoid and challenging to pipette, requiring vigorous vortexing.” – Line 209-211 of the revised manuscript.

We have also added a statement highlighting that insufficient sample volume did not impact sensitivity in our study: “…, though this did not appear to impact sensitivity.” – Line 205.

Lastly, we have also commented on the clinical characteristics of the discordant results: “Patient demographic or clinical presentation at the time of sample collection did not appear to impact test outcome between sample types in this study.” – Lines 201-202.

Figure 1 may be easier to read as an X-Y scatterplot, with Ct swab on the X axis and Ct Saliva on the X axis, and a diagonal line where they are equal from the bottom left to top right. The Ct cutoffs for positivity could also be indicated on such a plot, as a vertical and horizontal line across the plot space.

Figure 1 has been amended as recommended by the reviewer.

  1. Currently Figure 1 only contains the concordantly positive samples, but the discordant samples should also be plotted on such a scatterplot, and can be indicated with a different color of point. Please also include the negative samples, with NAs plotted at the maximum values on the plot. This will help to evaluate the extent of difference between positive and negative samples using the two different assays, which is important for evaluating the reliability of the saliva approach.

Figure 1 has been amended as recommended by the reviewer.

  1. Were any samples tested repeatedly on the same day (technical replicates)? This would serve as a valuable baseline to evaluate the delta values in Table 3, to see if the delta Ct values are within the range expected for technical replicates. It seems as if there is some amount of technical noise, since in many cases, there is more amplification following storing for 3 or 7 days.

Though intra-assay variability was assessed as part of a technical validation to establish assay parameters, this was not conducted using the clinical samples obtained in this study. However, we have added a statement to address the decreasing Ct values observed in saliva samples, which has also been observed by other investigators. “Interestingly, decreased Ct was observed in some donor samples during sample stability studies, suggesting a paradoxical increase in SARS-CoV-2 RNA. This effect has been observed in other evaluations of SARS-CoV-2 stability in saliva and is thought to be due to assay variability or possibly even virus replication in residual host cells within the sample [5]; however, this effect was not explored further as part of this study.” – Line 176-180 of the revised manuscript.

Reviewer 2 Report

The authors test the presence of SARS-CoV-2 virus RNA in saliva samples from a large number of patients that are detected by the classical method using NPS. They only identify 7 discrepancies. The study is well carried out. However, the presence of mucus or the low volume can make it difficult to standardize the technique. Additionally, 8.8% have to repeat the test, with the cost that this might imply.

Specific comments:

In the results section (page 3, line 121) it is specified that there are 3 invalid samples. Does this mean that the results of 429 samples are finally analyzed? If so, the results in Table 1 should be modified.

Table 2 shows the results of the discordant ones; however I do not understand the results of 2 of the samples. The CT values ​​in sample 2 are consistent with a negative result for NPS; however it is displayed as positive. In the case of sample 5, the table indicates that the result is positive in saliva and the Ct value is 37. Could you specify what are the targets 1 and 2? Why does sample 5 have no results for target 1? An explanatory legend is missing from this table.

A more detailed section explaining the saliva detection method would be appreciated by the reader.

Is there a relationship between the discordant samples and the initial volume of the sample or the amount of mucus present in it?

Author Response

  1. In the results section (page 3, line 121) it is specified that there are 3 invalid samples. Does this mean that the results of 429 samples are finally analyzed? If so, the results in Table 1 should be modified.

We agree that the current wording in this section does not align with the data presented in Table 1, since invalid results were excluded from this analysis. We have made the following revisions to clarify this point: “… and included in final analysis.” – Line 123 of the revised manuscript. “These three repeatedly invalid samples were excluded from final analysis.” – Line 127-128.

  1. Table 2 shows the results of the discordant ones; however I do not understand the results of 2 of the samples. The CT values ​​in sample 2 are consistent with a negative result for NPS; however it is displayed as positive. In the case of sample 5, the table indicates that the result is positive in saliva and the Ct value is 37. Could you specify what are the targets 1 and 2? Why does sample 5 have no results for target 1? An explanatory legend is missing from this table.

We have added a footnote to Table 2 to describe the assay targets used for SARS-CoV-2 detection. We have also modified the table to better indicate ‘dashed lines’ as ‘not detected’ targets. – Lines 142-146

All sample results as indicated in Table 2 are correct as per the manufacturer’s instructions for interpretation or by using validated thresholds. Although the obtained Ct values are high, they are considered within the assay cutoff values and are considered as positive or detected results.

  1. A more detailed section explaining the saliva detection method would be appreciated by the reader.

We have added more detail to the Roche cobas SARS-CoV-2 assay: “The cobas® SARS-CoV-2 test provides fully automated sample preparation (nucleic acid extraction and amplification) and RT-PCR-based qualitative detection of SARS-CoV-2 RNA through amplification of two genomic target regions, ORF1ab and E gene. An armored RNA internal control is added to all samples to validate each reaction. Automated data management software assigns test results for all tests. This test is Health Canada approved for in vitro diagnostic use.” – Lines 87 to 93 of revised manuscript.

  1. Is there a relationship between the discordant samples and the initial volume of the sample or the amount of mucus present in it?

With regards to discordant samples, there was one highly viscous saliva sample that was falsely-negative when compared to NPS testing. There were no other notable differences between paired samples discordant for SARS-CoV-2 detection in terms of clinical presentation or time from collection to testing. We have amended the manuscript to include a statement regarding the single false-negative saliva that was highly viscous: “Importantly, one discordant saliva sample that was falsely-negative compared to the matched NPS-collected sample was noted to be highly mucoid and challenging to pipette, requiring vigorous vortexing.” – Line 209-211 of the revised manuscript.

We have also added a statement highlighting that insufficient sample volume did not impact sensitivity in our study: “…, though this did not appear to impact sensitivity.” – Line 205.

Lastly, we have also commented on the clinical characteristics of the discordant results: “Patient demographic or clinical presentation at the time of sample collection did not appear to impact test outcome between sample types in this study.” – Lines 201-202.

Round 2

Reviewer 1 Report

The authors have addressed the reviewer concerns. 

However, looking at the revised Figure 1, there are points along the top and the right hand side that appear to be discordant but are not colored that way or indicated in Table 2. For example, there appears to be one sample with a value of ~16CT for nasopharyngeal, but at the maximum value for saliva. Why is this one not considered "discordant" between the two? One would expect that all of them along the top and right side would be considered discordant. 

Author Response

  1. However, looking at the revised Figure 1, there are points along the top and the right hand side that appear to be discordant but are not colored that way or indicated in Table 2. For example, there appears to be one sample with a value of ~16CT for nasopharyngeal, but at the maximum value for saliva. Why is this one not considered "discordant" between the two? One would expect that all of them along the top and right side would be considered discordant.

There were six positives samples for which a corresponding Ct value for either saliva, NPS, or both samples, could not be retrieved. These values were displaying incorrectly on the figure for only the sample where Ct value was present. We have removed any value that is not present for both sample types from the figure.

Reviewer 2 Report

I still believe that Table 1 should not contain information from 432 patients, since as well indicated in the results section, there are 3 patient samples that are not included in the final analysis. Therefore, this table should only contain the information of 429 patients since I understand that at least the saliva sample could not be analyzed. Are these three invalid samples within the discordant ones?

Author Response

  1. I still believe that Table 1 should not contain information from 432 patients, since as well indicated in the results section, there are 3 patient samples that are not included in the final analysis. Therefore, this table should only contain the information of 429 patients since I understand that at least the saliva sample could not be analyzed. Are these three invalid samples within the discordant ones?

The three samples that repeatedly tested as invalid were incorrectly included in Table 1 as negative results. These have already been excluded from analysis and Table 1 has been updated to reflect this change.